# Switching from Apoptosis to Pyroptosis: Gasdermin-Elicited Inflammation and Antitumor Immunity

**DOI:** 10.3390/ijms22010426

**Published:** 2021-01-04

**Authors:** Kohsuke Tsuchiya

**Affiliations:** 1Division of Immunology and Molecular Biology, Cancer Research Institute, Kanazawa University, Kakuma-machi, Kanazawa 920-1192, Japan; ktsuchiya@staff.kanazawa-u.ac.jp; Tel.: +81-76-264-6721; 2Institute for Frontier Science Initiative (InFiniti), Kanazawa University, Kakuma-machi, Kanazawa 920-1192, Japan

**Keywords:** pyroptosis, gasdermin, immunogenic cell death, antitumor Immunity, tumor microenvironment

## Abstract

Pyroptosis is a necrotic form of regulated cell death. Gasdermines (GSDMs) are a family of intracellular proteins that execute pyroptosis. While GSDMs are expressed as inactive forms, certain proteases proteolytically activate them. The *N*-terminal fragments of GSDMs form pores in the plasma membrane, leading to osmotic cell lysis. Pyroptotic cells release pro-inflammatory molecules into the extracellular milieu, thereby eliciting inflammation and immune responses. Recent studies have significantly advanced our knowledge of the mechanisms and physiological roles of pyroptosis. GSDMs are activated by caspases and granzymes, most of which can also induce apoptosis in different situations, for example where the expression of GSDMs is too low to cause pyroptosis; that is, caspase/granzyme-induced apoptosis can be switched to pyroptosis by the expression of GSDMs. Pyroptosis appears to facilitate the killing of tumor cells by cytotoxic lymphocytes, and it may also reprogram the tumor microenvironment to an immunostimulatory state. Understanding pyroptosis may help the development of cancer immunotherapy. In this review article, recent findings on the mechanisms and roles of pyroptosis are introduced. The effectiveness and limitations of pyroptosis in inducing antitumor immunity are also discussed.

## 1. Introduction

Previously, cell death was classified into two types: apoptosis and accidental necrosis, but recently, several forms of regulated necrosis have been discovered and received considerable interest [1,2]. Accidental necrosis is passive cell death caused by, for example, mechanical, thermal, and chemical damage to the cell membrane. On the other hand, regulated cell death, including apoptosis and regulated necrosis, is caused by a genetically encoded machinery, in other words, it is “programmed”.

Apoptosis is characterized by cell shrinkage, membrane blebbing, phosphatidylserine externalization, nuclear DNA fragmentation, and nuclear condensation [1,2]. Moreover, cell membrane integrity is preserved in the early stages of apoptosis, and apoptotic cells are efficiently cleared from tissues by phagocytes before becoming lytic. This is important for avoiding unwanted inflammation and autoimmunity, which are attributed to the release of pro-inflammatory molecules and self-antigens from dying cells. For these reasons, apoptosis is considered to be immunologically silent and even anti-inflammatory, though its pro-inflammatory effects have also been described in some settings [1,2]. Caspases are a family of intracellular cysteine proteases, and the majority of them participate in the induction of apoptosis [2,3,4]. Caspase-8 and caspase-9 are activated in death receptor-mediated (also known as extrinsic) and intrinsic apoptosis pathways, respectively. These caspases, in turn, serve as initiator caspases that directly or indirectly activate downstream effector caspases, such as caspase-3 and caspase-7. Effector caspases then cleave a number of substrates, including Rho-associated protein kinase I, ATPase 11A/C, Xk-related protein 8, and inhibitor of caspase-activated DNase. The cleavage of so-called death substrates leads to apoptosis with the above-mentioned properties [2,3].

In contrast to apoptosis, regulated necrosis occurs with rapid plasma membrane damage [1]. Regulated necrosis is considered to be inflammatory, because cells undergoing regulated necrosis rapidly release pro-inflammatory intracellular contents, including damage-associated molecular patterns (DAMPs) and cytokines. Several modes of regulated necrosis have been reported, for example, pyroptosis, necroptosis, and ferroptosis, which are induced by different stimuli through different signal transduction pathways [1,2]. While necroptosis is mediated by the pseudokinase mixed lineage kinase domain-like (MLKL), pyroptosis is mediated by gasdermin (GSDM) family proteins. Ferroptosis involves abnormalities in lipid peroxidation metabolism and iron homeostasis. In necroptosis, MLKL is activated by phosphorylation of key residues and subsequently act as an effector of plasma membrane permeabilization, leading to osmotic cell swelling and eventual necrotic lysis. On the other hand, in pyroptosis, GSDM proteins are proteolytically activated by certain proteases and subsequently form plasma membrane pores, which also lead to cell swelling and lysis [4,5].

Recent studies have identified proteases that activate GSDMs. There have also been new findings about the mechanisms by which pyroptosis induces inflammation and its impact on the tumor microenvironment. This review article introduces recent findings on the mechanisms and roles of pyroptosis and discusses its effectiveness and limitations in cancer treatment.

## 2. The Characteristics and Mechanism of Pyroptosis

Pyroptosis was originally proposed by Cookson and Brennan as caspase-1-dependent non-apoptotic cell death induced in macrophages during infection with *Salmonella* Typhimurium [6,7]. In 2001, these researchers coined the term “pyroptosis” that stems from the Greek roots “pyro”, relating to fire or fever, and “ptosis” to denote a falling, to describe pro-inflammatory regulated necrosis [7]. Later, human caspase-4 and caspase-5 and their mouse ortholog, caspase-11, have also been demonstrated to induce pyroptosis [8,9,10]. Pyroptotic cells rapidly lose cell membrane integrity, increase in size, and have smaller nuclei [4,5,11]. DNA damage also occurs upon pyroptosis, and pyroptotic cells become positive in terminal deoxynucleotidyl transferase-mediated dUTP nick-end labeling with a lower intensity than apoptotic cells. Intracellular pro-inflammatory molecules are rapidly and efficiently released from pyroptotic cells, thus triggering inflammation.

Although the mechanism of pyroptosis had long been unclear, in 2015, GSDMD, a 53 kDa cytoplasmic protein, was identified as a critical mediator of pyroptosis induced by inflammatory caspases (caspase-1, -4, -5, and -11) [12,13]. These caspases proteolytically activate GSDMD, leading to pore formation in the plasma membrane and pyroptosis (Figure 1). GSDMD consists of an *N*-terminal pore-forming domain, a *C*-terminal regulatory domain, and a central linker region. The pore-forming activity of the *N*-terminal domain is inhibited by the *C*-terminal domain in full-length GSDMD [14,15]. The cleavage of GSDMD by inflammatory caspases at the linker region liberates the *N*-terminal domain from the *C*-terminal domain. The *N*-terminal fragment (GSDMD-N) binds to phospholipids on the inner leaflet of the plasma membrane and in the mitochondria, such as phosphatidylinositol phosphates, phosphatidic acid, phosphatidylserine, and cardiolipin. GSDMD-N molecules, in turn, undergo conformational changes that facilitate oligomerization and membrane insertion to form transmembrane β-barrel pores with an inner diameter of 10–15 nm [16,17]. The plasma membrane pores cause water influx driven by oncotic pressure, cell swelling, and, ultimately, cell lysis [11].

The formation of GSDMD pores in the plasma membrane allows the influx of Ca^2+^ from the extracellular milieu, which directs calpain activation and the assembly of the endosomal sorting complex required for transport (ESCRT) machinery during pyroptosis (Figure 1) [18,19]. The Ca^2+^-dependent proteases calpains promote severe rupture of pyroptotic cell membranes by degrading vimentin intermediate filaments [18]. This is required for the release of macromolecules and organelles from pyroptotic cells. On the other hand, the ESCRT machinery promotes budding and pinching off of the damaged membrane, thereby mediating plasma membrane repair. Depletion of the ESCRT-III complex enhanced pyroptosis, suggesting that ESCRT-dependent membrane repair controls this cell death [19]. Therefore, GSDMD pore-mediated Ca^2+^ influx has opposite effects on pyroptosis, which may depend on the stage of the cell death process. Taken together, GSDMD pores not only execute pyroptosis, but also regulate it positively and negatively, and the fate of a cell, in which GSDMD is activated, may be determined at least in part by balance between the amount of GSDMD pores and the Ca^2+^-dependent regulation mechanisms.

Recent studies have revealed the crystal structures of full-length GSDMs and complexes between inflammatory caspases and GSDMD and the cryo-electron microscopy structure of a GSDMA3 oligomer [16,17,20,21]. The dynamics of pore formation by GSDMD-N has also been visualized by atomic force microscopy [22,23]. The structural insights may guide future drug design targeting GSDMs for controlling or facilitating pyroptosis.

## 3. GSDM Family Proteins

The human and mouse genomes encode six (GSDMA, GSDMB, GSDMC, GSDMD, GSDME, DFNB59) and ten (GSDMA1, GSDMA2, GSDMA3, GSDMC, GSDMC2, GSDMC3, GSDMC4, GSDMD, GSDME, DFNB59) members of GSDM family proteins, respectively [24,25]. While GSDME and DFNB59 are conserved in diverse species from fishes to mammals, other GSDM family genes are only conserved in birds to mammals. All the members of the GSDM family (except DFNB59) consist of two domains that are connected by a central linker region. The *N*-terminal domains of the GSDM proteins as well as that of GSDMD have the ability to form plasma membrane pores, resulting in cell lysis [14,25]. Hence, like GSDMD, other GSDM members have been expected to execute pyroptotic cell death after being processed by proteases at the linker region. Indeed, recent studies have shown that proteolytic activation of GSDME, GSDMB, and GSDMC by certain caspases and granzymes can lead to necrotic cell death (Figure 2) [25,26,27,28,29,30]. The Nomenclature Committee on Cell Death has proposed to define pyroptosis as depends on the formation of plasma membrane pores by members of the GSDM family, often (but not always) as a consequence of inflammatory caspase activation [1]. It has also been demonstrated that GSDM members can be cleaved by multiple proteases that activate or inactivate them, depending on the cleavage site. Of note, most of the proteases that induce pyroptosis can also induce apoptosis in the absence of the corresponding GSDM protein, which means that GSDMs can convert apoptosis into pyroptosis.

### 3.1. GSDMD

GSDMD was the first identified pyroptosis executor that acts downstream of inflammatory caspases [12,13]. Caspase-1 is activated in inflammasomes, cytosolic multiprotein oligomers formed in response to microbial and sterile stimuli [31,32]. An inflammasome complex comprises a pattern recognition receptor (PRR), which defines the inflammaosme, and the inactive zymogen pro-caspase-1. In inflammasomes, pro-caspase-1 molecules are brought into close proximity, resulting in the activation of caspase-1, which in turn cleaves GSDMD to induce pyroptosis. Active caspase-1 also processes the biologically inactive pro-cytokines, pro-interleukin (IL)-1β and pro-IL-18, into the active mature forms [31,32]. These cytokines lack a signal sequence for secretion and are released from cells by unconventional mechanisms, such as the formation of plasma membrane pores by GSDMD-N [33,34]. Importantly, caspase-1 initiates pyroptosis only in cells expressing sufficiently high levels of GSDMD to cause pyroptosis, including macrophages, monocytes and dendritic cells (DCs). On the other hand, caspase-1 initiates apoptosis in cell types that do not express sufficient levels of GSDMD, such as neuronal cells and mast cells [5,35,36]. Bid and caspase-7 are caspase-1 substrates. tBid, the *C*-terminal fragment of caspase-1-cleaved Bid, activates the mitochondrial apoptosis pathway, and Bid is the major mediator of caspase-1-induced apoptosis [35]. Caspase-7 plays a complementary role in caspase-1-induced apoptosis, as caspase-7 contributes to cell death when Bid is depleted [37]. In a previous study, UVB-irradiated human primary keratinocytes underwent caspase-1-dependent apoptosis, most likely because of inefficient activation of GSDMD [38]. Degradation of the major vault protein by caspase-1 might be involved in the induction of apoptosis. In neutrophils, GSDMD-N targets azurophilic granules, but not the plasma membrane, and consequently does not cause pyroptosis after inflammasome activation [39].

Caspase-11/4/5 directly bind with lipopolysaccharide, resulting in their activation via dimerization-induced autoproteolysis, without the need for inflammasome-forming PRRs [9,10]. Thus, these caspases serve as sensors of cytosolic LPS. LPS may be released into the cytoplasm during infection with Gram-negative bacteria, which enables it to gain access to caspase-11/4/5 [40]. Outer membrane vesicles and high mobility group box-1 protein (HMGB1) can also deliver LPS into the cytoplasm [41,42]. Active caspase-11/4/5 induces pyroptosis by cleaving GSDMD at the same site as caspase-1 [12,13]. Caspase-11/4/5 are considered to contribute to the pathology of endotoxemia through the induction of pyroptosis. Activation of caspase-11 in neutrophils appears to induce GSDMD-dependent extrusion of neutrophil extracellular traps [43]. Caspase-11 has been suggested to recognize not only LPS, but also endogenous and protozoan ligands [44,45].

In addition to inflammatory caspases, caspase-8, neutrophil elastase, and cathepsin G have also been demonstrated to proteolytically activate GSDMD. The caspase-8-GSDMD pathway operates in pyroptosis induced by *Yersinia* infection [46,47]. In aged neutrophils, neutrophil elastase released from granules into the cytoplasm induces cell death by activating GSDMD, which negatively regulates neutrophil accumulation and inflammation in the site of bacterial infection [48]. The intracellular serine protease inhibitors (serpin) B1a and serpin B6a protect neutrophils from cathepsin G-mediated cell death. In *Serpinb1a*/*Serpinb6a*-double mutant neutrophils, GSDMD is cleaved by cathepsin G at the linker region, resulting in increased inflammation, whereas GSDMD is dispensable for the neutrophil cell death [49]. In contrast, caspase-3 cleaves GSDMD within the *N*-terminal domain, which inactivates the cytolytic activity of GSDMD [26,50].

During infections, pyroptosis may eliminate intracellular replication niches for intracellular parasitic pathogens [51]. Pyroptosis may also promote the recruitment and activation of leukocytes through the release of DAMPs and IL-1 family cytokines. GSDMD forms pores not only in the membranes of eukaryotic cells, but also in those of bacterial cells, which enables it to kill bacteria directly [15,52]. During pyroptosis of host macrophages, intracellular bacteria may be released into the extracellular space, where neutrophils can kill the released bacteria, or may be trapped in the pyroptotic cell corpse, termed pore-induced intracellular traps (PITs), which prevents the dissemination of pathogens [51,53]. PITs may also be engulfed (efferocytosed) by macrophages and neutrophils to kill the bacteria trapped inside. GSDMD has been shown to contribute to host defense against bacterial and viral pathogens, including *Burkholderia thailandensis*, *Brucella abortus*, and rotavirus, as GSDMD-deficient mice were more susceptible to these pathogens than control mice [52,54,55]. In *Paracoccidioides brasiliensis* infection, caspase-11-and GSDMD-dependent pyroptosis occurs and promotes IL-1α release, which protects the host by inducing the production of nitric oxide and IL-17 [56]. Mice lacking both GSDMD and caspase-7 showed decreased resistance to *Legionella pneumophila* compared with control mice, while mice lacking either of them did not, suggesting that GSDMD-mediated pyroptosis and caspase-1/8-induced apoptosis are functionally redundant in restricting *L. pneumophila* infection [57]. GSDMD has also been implicated in the pathogenesis of autoinflammatory and autoimmune diseases, such as cryopyrin-associated periodic syndromes, familial Mediterranean fever, experimental autoimmune encephalomyelitis, and graft-versus-host disease following allogeneic hematopoietic stem cell transplantation [58,59,60,61].

### 3.2. GSDME

GSDME (also known as DFNA5) was identified as the causative gene for nonsyndromic hearing loss and has been considered as a tumor suppressor [24,62]. GSDME is proteolytically activated by caspase-3, and the resultant *N*-terminal fragment, like GSDMD-N, induces pyroptosis [26,27]. Caspase-3 is known to play a central role in morphological changes and execution of apoptosis [1,2]. While apoptosis is not a lytic form of cell death, cells expressing high levels of GSDME rapidly undergo pyroptosis after caspase-3 activation. GSDME is expressed in a wide range of tissues and cell types, particularly in the central nervous system, small and large intestine, and reproductive tissues. On the other hand, GSDME expression is silenced in most cancer cells, likely attributed to promoter methylation of the *GSDME* gene [63,64,65,66]. Indeed, GSDME expression in cancer cells was upregulated after treatment with the DNA methyltransferase inhibitor 5-Aza-2’-deoxycytidine (decitabine). Since the silencing of the *GSDME* gene makes it difficult to induce cancer cell pyrotosis with apoptosis-inducing drugs, and since reduced expression levels of GSDME are associated with poor prognosis in cancer patients, the promoter methylation may be a potential drug target in cancer therapy [30,64]. However, chemotherapy drug-induced toxicity in normal tissues also involves GSDME-mediated pyroptosis, indicating that attempts to increase GSDME expression, for example by DNA methyltransferase inhibition, should be combined with cancer-targeted drug delivery and/or molecular-targeted anticancer drugs to avoid the detrimental effects of GSDME [27]. At the terminal stage of apoptosis, cells become necrotic, called secondary necrosis [2]. Given that caspase-3 can induce pyroptosis via GSDME maturation, the question arises whether GSDME is a mediator of secondary necrosis. GSDME causes rapid cell lysis (pyroptosis) upon treatment with apoptotic stimuli in cells expressing it at high levels. However, loss of membrane integrity eventually occurs in late apoptosis, even in the absence of GSDME [67]. If secondary necrosis is defined as passive lysis of cells in late apoptosis, it can be distinguished from GSDME-mediated pyroptosis, which is active cell lysis.

Granzyme B is a serine protease found in and released from cytotoxic granules of cytotoxic T lymphocytes (CTLs) and natural killer (NK) cells that recognize infected and cancerous cells to eliminate them as target cells [68,69]. CTLs and NK cells interact with the target cells to elicit specific killing, upon which granzyme B released from cytotoxic granules diffuses into the cytoplasm of the target cells via plasma membrane pores formed by perforin. Granzyme B in the cytoplasm induces apoptosis, as it proteolytically activates Bid and caspase-3, degrades the anti-apoptotic protein Mcl-1, and can cleave other key caspase substrates [68,69]. Granzyme B can also enter into the mitochondria to induce reactive oxygen species (ROS)-dependent cell death through the disruption of the mitochondrial respiratory chain complex I [70]. A recent report has demonstrated that granzyme B cleaves GSDME at the linker region, leading to pyroptosis [30]. In Hela cells overexpressing GSDME, pyroptosis occurred after incubation with the human NK cell line YT cells. The induction of pyroptosis was in part independent of caspase-3, as it was only partially reduced in the absence of caspase-3. The ability to evade apoptosis is a hallmark of cancers and mediated by downregulation or defective function of pro-apoptotic molecules and upregulation of anti-apoptotic molecules. However, the direct activation of GSDME by granzyme B implies that CTLs and NK cells are capable of inducing pyroptosis in GSDME-expressing cancer cells even when apoptosis signaling pathways are impaired in the targets.

GSDME is conserved in fishes. A teleost fish orthologue of GSDME is activated efficiently by caspase-1 and, to a lesser extent, by caspase-3/7, to mediate pyroptosis [71]. Besides, zebrafish has two GSDME orthologues, GSDMEa, and GSDMEb, which are predicted to be cleaved by caspase-3 and caspase-1, respectively [72]. Hence, lower vertebrate GSDME may serve a function analogous to that of mammalian GSDMD.

### 3.3. GSDMB

Human GSDMB has at least four isoforms (isoform 1-4) with different linker region sequences. All of the hGSDMB isoforms have recently been described to be cleaved and activated by granzyme A, which, as well as granzyme B, is abundant in cytotoxic granules [28,68]. Cytotoxic lymphocytes could induce pyroptosis in their target cells expressing sufficient levels of GSDMB. Granzyme A also processes the electron transport chain components and the SET complex components, resulting in ROS generation and single stranded DNA nicks, thereby inducing caspase-independent cell death with apoptotic morphology [70,73]. Thus, GSDMB joined the death substrates for granzyme A, while it mediates pyroptosis. GSDMB is expressed in normal tissues, including the gastrointestinal mucosa, esophageal epithelium, tongue, trachea, and bladder [28]. GSDMB is often expressed in cancer cells derived from these tissues, whereas it may be silenced in esophageal and gastric cancer cells. Besides, significant correlations were found between expression levels of GSDMB and overall survival of patients with bladder carcinoma or skin cutaneous melanoma [28]. Strikingly, GSDMB expression is increased in cancer cells stimulated with cytokines, including interferon (IFN)-γ and tumor necrosis factor (TNF)-α, which makes us speculate that cytokine production and cytotoxic mechanisms cooperate in inducing GSDMB-mediated pyroptosis in cancer cells [28]. However, GSDMB is overexpressed in several kinds of cancers and has also been proposed to exert pro-tumor activities, though how this protein promotes tumor progression remains unclear [74,75,76].

Single nucleotide polymorphisms (SNPs) in the *GSDMB* gene were associated with increased risks of asthma, Crohn’s disease, inflammatory bowel disease, and ulcerative colitis [77,78]. GSDMB was highly expressed in bronchial epithelial cells from asthmatic patients, and human GSDMB-expressing transgenic mice showed spontaneous increases in airway hyperresponsiveness, peribronchial smooth muscle, and collagen deposition. A previous report has proposed that caspase-1 proteolytically activates the longest isoform (isoform 3) of human GSDMB, but not GSDMB isoform 1 that lacks 13 amino acids in the interdomain linker encoded by exon 6 [79]. The SNPrs11078928, which prevents the splicing of exon 6, and thus abolishes the expression of GSDMB isoform 3, was associated with decreased asthma risk, implying that caspase-1-induced, GSDMB-mediated pyroptosis perhaps plays a role in asthma pathogenesis. However, in other studies, GSDMB was not activated by caspase-1 [78,80]. Rather, caspase-3, 6, and 7 cleaved GSDMB within the N-terminal domain, resulting in inactivation of the pore-forming activity. Our unpublished data also do not support the idea that caspase-1 directly activates GSDMB. In our experiments, GSDMD-deficient THP1 cells, which did not undergo pyroptosis after LPS priming followed by stimulation with the inflammasome activator nigericin, were transduced with full-length human GSDMD, GSDMB isoform 1, or GSDMB isoform 3. Pyroptosis was induced by LPS plus nigericin in the GSDMD-deficient cells complemented with GSDMD, but not in those with the GSDMB isoforms (data not shown). Moreover, both GSDMB isoforms were cleaved within the *N*-terminal domain following inflammasome activation, most likely due to caspase-1/8-induced activation of executioner caspases (data not shown). Hence, these results are consistent with those in the latter studies.

Unlike other GSDMs, GSDMB can bind to phospholipids even at the intact form (full-length protein), whereas it requires proteolytic maturation for inducing pyroptosis [78]. In addition, GSDMB binds to sulfatide, but not cardiolipin. Another unique property of GSDMB is the ability to enhance the cleavage of GSDMD by caspase-4 [80]. GSDMB directly interacted with the CARD domain of caspase-4 through its *N*-terminus, which might augment the enzymatic activity of caspase-4.

### 3.4. GSDMC

GSDMC was first identified as a marker for melanoma progression [81]. GSDMC is expressed in the proximal digestive tract, skin, spleen, tonsil, and female tissues. In a murine colon cancer model, expression of *Gsdmc2*/*Gsdmc4* was increased by blockade of transforming growth factor (TGF)-β signalling [82]. Upregulation of *GSDMC* was found in human colorectal cancer tissues, and its oncogenic potential was experimentally confirmed [82]. In human keratinocytes, UV irradiation increased the expression of GSDMC, which in turn contributed to the expression of matrix metalloproteinase-1 [83].

A recent study has demonstrated that GSDMC is a substrate of and activated by caspase-8, an initiator caspase in the extrinsic apoptosis pathway, and caspase-6 [81]. In that study, the expression of GSDMC, but not other GSDMs, was increased in cancer cells by nuclear translocation of programmed death ligand 1 (PD-L1), which was induced by hypoxia or chemotherapeutic drugs. Treatment with TNF-α (plus cyclophosphamide) induces apoptosis via caspase-8 activation [2]. TNF-induced apoptosis was observed in cell lines of multiple cancer types under normoxia, whereas the same treatment induced necrotic cell death under hypoxia. The necrotic cell death was found to depend on caspase-8 and GSDMC, suggesting that increased GSDMC expression under hypoxic conditions can switch caspase-8-induced apoptosis to pyroptosis. Necrotic areas are commonly found in central hypoxic regions of solid tumors. The activation of the caspase-8-GSDMC pathway by TNF-α derived from macrophages may account at least in part for tumor necrosis in hypoxic regions, as blocking this signaling pathway drastically reduced areas of necrosis in cancer xenografts in vivo. Moreover, although various types of chemotherapy drugs could increase the expression of GSDMC, only antibiotic chemotherapy drugs, including doxorubicin, daunorubicin, epirubicin, and actinomycin D, induced GSDMC expression and caspase-8 activation simultaneously, causing GSDMC cleavage followed by pyroptotic cell death. Hence, it may be speculated whether an anticancer drug induces GSDMC-mediated pyroptosis or apoptosis is determined by whether it can activate caspase-8.

### 3.5. GSDMA

The protease that activates GSDMA has not been discovered to date. Mouse GSDMA1 was the first cloned member of the GSDM family and was named for its expression pattern [84]. GSDMA1 is expressed exclusively in the upper gastrointestinal tract, especially the stomach and skin. Human GSDMA is expressed in the epithelium of the stomach, esophagus, mammary gland, and skin, but is frequently silenced in gastric cancer cells [85]. GSDMA seemed to participate in TGF-β induction of gastric epithelial cell apoptosis. Mouse GSDMA3 is expressed in the hair follicle of skin. Although GSDMA3 appeared to be dispensable for the development of skin, gain-of-function mutations in this protein have been shown to cause hair loss and keratosis [86,87,88]. The gain-of-function mutations were found to impair the interaction between the *N*-terminal and *C*-terminal domains of GSDMA3, which disrupts the autoinhibition, leading to constitutive activation and pyroptosis induction without cleavage at the interdomain linker [88]. It was also shown that active GSDMA3 can promote autophagy and mitochondrial ROS production.

## 4. Pyroptosis-Elicited Inflammation

Pyroptosis has been considered inflammatory, as cells undergoing pyroptosis release pro-inflammatory molecules and organelles, such as nucleotides, IL-1 family cytokines, HMGB1, nucleic acids, mitochondria [5]. The formation of GSDM pores in the plasma membrane causes the dissipation of ion gradients and release of small cytosolic molecules that can pass through the pores. Thereafter, GSDM pores cause cell lysis due to water influx, allowing the release of large cytosolic contents. It is thus likely that DAMPs of different sizes may be released from pyroptotic cells at different times or stages (Figure 3). In the following subsections, individual DAMPs released from pyroptotic cells are described.

### 4.1. ATP

ATP is small enough to pass through GSDMD pores. ATP release occurred prior to cell lysis upon pyroptosis, suggesting that ATP is a DAMP that can be released through pores formed by GSDMs [35,89]. Apoptosis has also been described to cause ATP secretion via the hemichannel pannexin-1 [90]. However, pyroptotic cells seemed to release ATP more efficiently than apoptotic cells, and AMP rather than ATP was released after apoptosis [89,91]. The release of ATP following the activation of caspase-1 largely depended on GSDMD [35]. Although GSDMD-deficient cells underwent apoptosis after caspase-1 activation, ATP release was significantly reduced and delayed in the cells as compared to GSDMD-sufficient cells that underwent pyroptosis after caspase-1 activation. In cells with GSDM pores, rapid efflux of nucleotides may lead to depletion of cellular ATP/dATP and consequently suppress the mitochondrial pathway of apoptosis, which requires these nucleotides. Extracellular ATP is sensed by P2X and P2Y purinergic receptors and exerts immunostimulatory effects [92]. ATP acts as a find-me-signal that recruits monocytes, macrophages, and DCs [89,90]. ATP also activates the NLRP3 (nucleotide-binding oligomerization domain-like receptor family, pyrin domain-containing 3) inflammasome and induces IL-1β production in a P2X7-dependent manner [32,93]. This signaling pathway has been proposed to play an important role in T cell priming by DCs pulsed with dying tumor cells [94].

### 4.2. IL-1β

Pro-IL-1β is processed into the mature form by caspase-1, although other proteases, including caspase-8, neutrophil elastase, and proteinase- 3, have also been suggested to proteolytically activate this cytokine [31,33]. Mature IL-1β has a wide spectrum of biological activities. For example, it causes fever, activates leukocytes, mediates transmigration of leukocytes, and promotes cell survival and proliferation. IL-1β lacks a signal sequence for secretion and is released from cells by unconventional mechanisms. During inflammasome-induced pyroptosis, IL-1β is released via GSDMD-formed pores without requiring cell membrane rupture, as inhibition of cell lysis did not affect the release of this cytokine [34]. The size of mature IL-1β fits through a GSDMD pore, possibly enabling the efficient efflux during pyroptosis. On the other hand, pro-IL-1β seemed to form a molecular complex, which might retain the unprocessed form of IL-1β inside pyroptotic cells [95].

Moreover, in several cases, mature IL-1β was released from living cells via GSDMD pores, termed hyperactivation [44,96,97]. For example, peptidoglycan-induced atypical NLRP3 inflammasome activation was followed by IL-1β release through GSDMD-formed pores, but not by cell death [97,98]. Oxidized phospholipids, such as oxidized 1-palmitoyl-2-arachidonoyl-sn-glycero-3-phosphorylcholine, have also been suggested to serve as hyperactivating ligands for caspase-11 [44]. Hyperactivation led to continuous release of IL-1β that lasted for several days, which may potentiate adaptive immune responses [96]. In line with this notion, a recent study demonstrated that DCs in the hyperactivation state induced strong protective immunity against tumors [99]. The abundance of GSDMD pores formed in hyperactivated cells may be too low to cause pyroptosis. The cellular mechanisms controlling pyroptosis, such as ESCRT-III-mediated membrane repair, may also contribute to the hyperactivation state. IL-1β release without cell death was also observed in macrophages stimulated with mycoplasmal lipoproteins/lipopeptides [100]. Interestingly, in that case, IL-1β release from living macrophages was independent of GSDMD, but rather through changes in membrane permeability.

In addition to macrophages, neutrophils are also a major source of IL-1β. It has recently been suggested that neutrophils secrete mature IL-1β through a non-lytic, autophagy-dependent mechanism [39]. Inflammasome-induced IL-1β release from neutrophils was significantly diminished when the autophagy conjugation enzyme ATG7 was depleted, whereas the cytokine release was increased by amino acid starvation, a well-described trigger of autophagy. GSDMD was also required for IL-1β release from neutrophils following canonical NLRP3 inflammasome activation. However, even though GSDMD-N was generated, the activated neutrophils did not undergo pyroptosis. GSDMD-N was localized to azurophilic granules and autophagosomes rather than the plasma membrane, which may be the reason why neutrophils are resistant to pyroptosis. It remains unclear how GSDMD promotes autophagy-dependent IL-1β secretion by neutrophils.

### 4.3. HMGB1

HMGB1 is a 25 kDa nuclear protein composed of two DNA-binding domains, boxes A and B, and a negatively charged C-terminal tail [101,102]. HMGB1 is released from cells in response to inflammatory stimuli and upon cell death. After inflammasome activation in vitro, HMGB1 was released in a GSDMD-dependent manner, and cell lysis was required for the release of HMGB1 from pyroptotic cells, indicating that GSDMD pores are not sufficient for it [103]. Extracellular HMGB1 has been described to bind to more than 10 different receptors, including Toll-like receptors (TLRs) and the receptor for advanced glycation end products (RAGE), to exert chemoattractant and cytokine-like activities [101,102]. Post-translational modifications of HMGB1 determine its subcellular localization and pro-inflammatory activities. Nuclear translocation of HMGB1 is prevented by hyperacetylation of lysine residues in nuclear localization sequences, resulting in cytoplasmic accumulation of hyperacetylated HMGB1. HMGB1 has three cysteine residues at positions 23, 45, and 106, and their redox state is critical for their immune activities [104,105,106]. Fully reduced HMGB1, in which all the cysteine residues are in the thiol form, forms a hetero-complex with C-X-C motif chemokine ligand 12 (CXCL12), which has stronger chemotactic activity than CXCL12 alone. On the other hand, HMGB1 with C106 in the thiol form and a disulfide bond between C23 and C45 (disulfide HMGB1) exerts cytokine-stimulating activity in a TLR4-dependent manner. Furthermore, terminal oxidation of any of the cysteines to sulfonates abrogates the ability of HMGB1 to induce cytokine production or chemotaxis [101,102]. It was shown that induction of pyroptosis alone caused the release of fully reduced HMGB1, whereas pyroptosis in combination with TLR2/4 stimulation led to that of both disulfide HMGB1 and fully reduced HMGB1 [107]. During apoptosis, HMGB1 is retained inside cells and exposed to high levels of ROS, leading to terminal oxidation of the critical cysteines to sulfonates. A study showed that fully oxidized HMGB1 plays a central role in apoptotic cell-induced immune tolerance [108]. Therefore, it appears that pyroptotic cells can release immunostimulatory forms of HMGB1 more efficiently than apoptotic cells.

Extracellular HMGB1 not only serves as a DAMP with immunomodulatory activities but also delivers LPS and immune-activating nucleic acids (RNA and DNA) into the cytoplasm [41,109]. HMGB1 binds with LPS and nucleic acids, and the complexes are endocytosed through binding to RAGE. After endocytosis, HMGB1 permeabilizes lysosomal membranes, which enables HMGB1-binding molecules to enter the cytoplasm. Then, LPS and nucleic acids gain access to their intracellular sensors and induce pyroptosis and cytokine production, respectively. All isoforms of HMGB1 tested (disulfide, sulfonyl, and fully reduced forms) were taken up by macrophages [110]. In a previous study, cytoplasmic delivery of extracellular LPS by hepatocyte-derived HMGB1 contributed to pyroptosis-dependent lethality in models of endotoxemia and bacterial sepsis [42].

### 4.4. Pore-Induced Breakdown of Ion Gradients

GSDMD pores in the plasma membrane allow non-selective flux of ions, such as K^+^ and Ca^2+^. Although caspase-11/4/5 do not process pro-IL-1β and pro-IL-18 directly, activation of these caspases can result in caspase-1 activation and subsequent maturation of these cytokines in a GSDMD-dependent manner [8,19]. K^+^ efflux via GSDMD pores occurs downstream of caspase-11/4/5, leading to the activation of the NLRP3 inflammsome, in which caspase-1 is activated [111]. The inflammasome formed by this mode is called the non-canonical inflammasome.

GSDMD pores cause extracellular Ca^2+^ influx, which in turn activates calpains. Calpains have been shown to degrade vimentin intermediate filaments in pyroptotic cells [18]. As a consequence of the disruption of the cytoskeleton, pyroptotic cells became susceptible to rupture by shear stress and compressive force. Severely ruptured pyroptotic cells could release pro-inflammatory/immunogenic macromolecules and organelles, such as mitochondrial DNA, mitochondria and nuclei. Apoptosis-associated, speck-like protein containing a caspase recruitment domain (ASC) is an inflammasome adaptor that forms highly cross-linked macromolecular protein complexes, called ASC specks, after inflammasome activation [31]. ASC specks can be released from pyroptotic cells [112]. Then, extracellular ASC specks can be taken up by phagocytes, in which the specks can recruit and activate caspase-1 to sustain and spread inflammasome-induced inflammation. The release of ASC specks was observed in severely ruptured pyroptotic cells, suggesting a role for calpains in the propagation of inflammasome responses [18]. IL-1α is a pro-inflammatory cytokine that shares the same receptor with IL-1β [33]. Although pro-IL-1α has cytokine activity, proteolytic maturation increases its potency and release from cells [113,114]. It has been known that IL-1α maturation occurs during inflammasome activation [115]. A recent study showed that IL-1α maturation induced by non-particulate inflammasome activators was mediated by Ca^2+^ influx via GSDMD pores and the resultant calpain activation [116]. Therefore, caspase-1 activation leads to the maturation of both IL-1α and IL-1β through different mechanisms, GSDMD pore formation and direct processing, respectively. IL-1α maturation by calpains may also be operative during pyroptosis caused by other GSDMs.

Pyroptosis-induced Ca^2+^ influx has also been demonstrated to mediate the generation of the lipid mediator eicosanoids [117]. Pyroptosis activates Ca^2+^-dependent phospholipase A_2_, which produces free arachidonic acid, thereby promoting the biosynthesis of eicosanoids, including prostaglandins (PG) and leukotrienes. Systemic inflammasome activation led to eicosanoid-dependent lethal inflammation. Eicosanoids are also involved in neutrophil recruitment to the PITs [118]. On the other hand, a study has suggested that PGE_2_ also serves as an inhibitory DAMP, as this lipid mediator in necrosis cell supernatants inhibited inflammatory responses to the supernatants [119].

Pyroptosis has been thought to contribute to the pathology of endotoxemia through the release of DAMPs (Figure 4) [8,19]. Recent studies have suggested that pyroptosis is also involved in the development of disseminated intravascular coagulation (DIC), which is associated with microvascular thrombosis, organ damage, and death [120,121]. GSDMD-dependent pyroptosis led to the release of microvesicles containing tissue factor (TF), a major initiator of the extrinsic coagulation (Figure 4). GSDMD and TF activity were required for systemic blood clotting, thrombosis in tissues, and lethality induced by systemic canonical or non-canonical inflammasome activation [120]. Interestingly, TF was activated owing to Ca^2+^ influx via GSDMD pores in pyroptotic cells. GSDMD-mediated Ca^2+^ influx was followed by the activation of transmembrane protein 16F (TMEM16F), a calcium-dependent phospholipid scramblase, resulting in the externalization of phosphatidylserine, which increases TF activity [121].

## 5. Pyroptosis and Antitumor Immunity

### 5.1. Immunogenic Cell Death, Apoptosis, and Pyroptosis

Immunogenic cell death (ICD) is defined as a form of regulated cell death that is sufficient to activate an adaptive immune response in immunocompetent hosts [1]. Certain chemotherapeutic drugs, radiotherapy, photodynamic therapy, and oncolytic viruses have been described to induce traditional ICD [122,123]. Cancer cells killed by ICD-inducing agents may elicit antigen-specific protective immunity against the cancer cells without any adjuvant. ICD of cancer cells in tumors may suppress tumor growth by eliciting anticancer immunity. Antigenicity and adjuvanticity are two key factors of ICD. Tumor-associated antigens that confer antigenicity may include antigens recognized by self-reactive low-affinity TCRs or may be derived from proteins with cancer cell-specific post-translational modifications, viral proteins, and peptides yielded by unconventional translation. Proteins with point mutations and frameshift mutations may also provide neoantigens that confer antigenicity to cancer cells. Upon ICD, dying cells release or expose DAMPs, which confer adjuvanticity, to promote DC maturation, antigen processing, and antigen presentation [122,123]. Well-studied ICD-associated DAMPs are calreticulin, ATP, HMGB1, nucleic acids, annexin-1, and type I IFNs produced by dying cells. Calreticulin, an endoplasmic reticulum (ER) resident protein, is translocated from the ER lumen to the outer surface of the plasma membrane during ICD [124]. Cell surface-exposed calreticulin functions as an eat-me-signal to promote phagocytosis by macrophages and DCs, resulting in cross-presentation of dead cell-associated antigens to CTLs. Calreticulin also signals through CD91 on immune cells, leading to the production of pro-inflammatory cytokines and Th17 cell priming [125]. Annexin-1 has been suggested to enhance the interaction between dying cancer cells and tumor-infiltrating DCs expressing formyl peptide receptor 1 (FPR1) [126]. In line with the role of these DAMPs in ICD, chemotherapy-elicited development of anticancer immunity has been demonstrated to require calreticulin, FPR1, P2X7, a receptor for ATP, and TLR4, a receptor for disulfide HMGB1 [94,124,126,127].

Unlike ICD, apoptosis is generally considered to be a less inflammatory and less immunogenic cell death process [2]. The low immunogenicity of apoptotic cells may be attributed to inefficient release of DAMPs and activation of apoptosis-related caspases, especially the executioner caspase-3 and caspase-7. During apoptosis, cleavage of flippase and scramblase by caspase-3/7 causes externalization of phosphatidylserine, which functions as an eat-me-signal to mediate clearance of apoptotic cells by phagocytes without inducing inflammation [2]. Caspase-3 also degrades cGAS and its downstream signaling molecules in apoptotic cells, thereby inhibiting type I IFN production induced by mitochondrial DNA released due to mitochondrial outer membrane permeabilization [128]. Besides, caspase-3 proteolytically activates cytosolic calcium-independent phospholipase A_2_, which leads to the production of PGE_2_, an immunosuppressive lipid product of arachidonic acid. Oxidization of HMGB1 may contribute to immune tolerance induced by apoptotic cells [108]. It was also demonstrated that apoptotic cells release AMP rather than ATP [91]. In that study, extracellular AMP was metabolized to adenosine, which then stimulated macrophages via the A2a adenosine receptor to induce the expression of anti-inflammatory genes.

Initially, ICD was thought to be an immunogenic type of apoptosis, since some of the ICD inducers are also known as inducers of apoptosis [122,123]. So far, regulated cell death is classified into apoptosis and non-apoptotic cell death, including regulated necrosis [1]. Given the pro-inflammatory properties of regulated necrosis, it is worth considering whether non-apoptotic cell death pathways described recently are involved in traditional ICD induction. Of note, in a recent study, GSDMC-dependent pyroptosis occurred in cancer cells after treatment with anthracyclines, such as doxorubicin and epirubicin that are well-known ICD inducers, but not with most other chemotherapy drugs tested [29]. These anthracyclines induced the expression of GSDMC and activation of caspase-8 simultaneously, resulting in proteolytic activation of GSDMC by caspase-8, whereas most other drugs only induced GSDMC expression. Hence, it is conceivable that cell death by some traditional ICD inducer may overlap with regulated necrosis. Pyroptosis, like ICD, rapidly leads to the release of ATP and immunostimulatory forms of HMGB1, but whether it causes calreticulin externalization is unclear [35,89,107]. Pyroptotic cells were efficiently taken up by professional and non-professional phagocytes [89,129]. Although phosphatidylserine was externalized upon pyroptosis, its involvement in phagocytosis of pyroptotic cells is debatable, as only one of the two phosphatidylserine-blocking proteins suppressed the phagocytosis [53,89,129]. Further research is needed to clarify the relationship between traditional ICD and pyroptosis.

### 5.2. Effects of Pyroptosis on the Tumor Microenvironment

Several recent studies have reported how pyroptosis impacts on the tumor microenvironment and anticancer immunity (Figure 5). As mentioned above, GSDMB is activated by granzyme A, while GSDME is activated by granzyme B and caspase-3. Expression of human GSDMB or GSDME sensitized cancer cells to cytotoxic lymphocyte-induced pyroptosis in a granzyme-dependent manner [28,30]. Since cell lysis will ensure death of target cells, pyroptosis probably enhances killing of cancer cells by cytotoxic lymphocytes. Moreover, overexpression of GSDME in melanoma and breast cancer cells significantly inhibited the growth of tumor xenografts in immunocompetent mice, while depletion of GSDME had the opposite effect [30]. The tumor-suppressive effect of GSDME relied on host CD8^+^ T cells, NK cells, and perforin, further supporting the importance of pyroptosis in lymphocyte cytotoxicity against tumor cells. Of note, it was demonstrated that tumor cell pyroptosis activates the tumor microenvironment toward an immunostimulatory state. Indeed, overexpression of GSDME in cancer cells significantly increased the number of intra-tumoral NK cells and antigen-specific CD8^+^ T, the expression of granzyme B, perforin, IFN-γ, and TNF-α in tumor-infiltrating lymphocytes (TILs), and phagocytosis of tumor cells by tumor-associated macrophages. Consistent results were found with melanoma xenografts treated with a molecular-targeted therapy [130]. Also, vaccination with GSDME-overexpressing cancer cells could confer resistance to subsequent challenge with the parental cancer cells [30]. Furthermore, Wang et al. showed with a novel technique (see the next paragraph) that pyroptosis induction in part of tumor cells led to T cell-dependent tumor regression accompanied with increased T cell, NK cell, M1 macrophage populations and decreased regulatory T cell, M2 macrophage, neutrophil, and myeloid-derived suppressor cell populations [131]. It is thus suggested that pyroptosis and cytotoxic lymphocytes can facilitate each other, forming a positive feedback loop in anticancer immunity. Given that GSDMB expression is upregulated by IFNs and TNF-α, cytokine production by TILs may also contribute to the positive feedback loop [28]. It is also noteworthy that IL-1β may play a critical role in pyroptosis-induced microenvironment activation, but it may depend on the case [30,131]. Cancer cells often express immune checkpoint molecules, which restrain T cell functions in tumors [132]. One example is PD-L1 that interacts with PD-1 on T cells to inhibit target recognition. Although neither blocking the PD-1-PD-L1 pathway nor transient pyroptosis induction inhibited 4T-1 tumor growth alone, the combination of them strongly suppressed the tumor growth [131]. Also, expression of human GSDMB in mouse colon carcinoma and melanoma cells did not affect the growth of tumors in immunocompetent mice. However, it markedly augmented the suppression of tumor growth achieved by immune checkpoint blockade with an anti-PD-1 antibody [28]. Accordingly, pyroptosis can function as a form of ICD that can synergize with immune checkpoint inhibition to elicit protective immune responses.

Whether chemotherapy drugs and cytotoxic lymphocytes induce pyroptosis or apoptosis in target cancer cells appears to depend on the expression of GSDMs in the target cells (Figure 2). The expression of GSDME, which acts as a tumor suppressor, is frequently silenced in cancer cells, most likely due to promoter methylation [63,64,65,66]. Thus, treatment with DNA methyltransferase inhibitors, such as decitabine, may restore GSDME expression, thereby sensitizing cancer cells to pyroptosis (Figure 6a). In line with this notion, cancer lines showed elevated GSDME expression upon decitabine treatment, and the cells underwent pyroptosis after treatment with the combination of decitabine and chemotherapeutics, including irinotecan, cisplatin, and tumor-targeting nanoliposome loaded with cisplatin (LipoDDP [65,133]. The combination therapy of decitabine and LipoDDP also inhibited the growth of 4T-1 xenografts and led to immune activation in the tumor microenvironment. Similar results were shown with decitabine, indocyanine green, and tumor-homing biomimetic nanoparticles loaded with them [134]. Considering that GSDME mediates chemotherapy drug-induced toxicity in normal tissues, these drug delivery systems may help to avoid the detrimental effects of GSDME. Another strategy to induce pyroptosis in tumor cells is to deliver active GSDM proteins into the cells. For that purpose, Wang et al. established a bioorthogonal chemical system, in which active GSDMA3 protein is conjugated to gold nanoparticles via a silyl ether linker that can be cleaved by desilylation that is catalyzed by phenylalanine trifluoroborate (Phe-BF3), a cancer-imaging probe (Figure 6b). Both the nanoparticles and Phe-BF3 selectively accumulate in tumors, enabling controlled release of active GSDMA3 in tumor cells [131]. In addition, viral vectors may also be employed to induce pyroptosis selectively in tumor cells. Schwannomas are peripheral nerve sheath tumors associated with severe disability. To induce pyroptosis in schwannomas, adeno-associated serotype 1 virus (AAV1)-based vector was engineered to express GSDMD-N under the control of the schwann-cell-specific promoter, P0 (Figure 6c) [135]. Intratumoral injection of the AAV1 vector expressing GSDMD-N suppressed the growth of schwannoma tumors and resolved tumor-associated pain without causing neurologic damage. Furthermore, caspase-1-dependent pyroptosis occurred in INR1G9 tumor cells upon magnetic intra-lysosomal hyperthermia generated by the combination of targeted magnetic nanoparticles and an alternating magnetic field through leakage of lysosomal cathepsins into the cytosol, providing a strategy to induce caspase-1 activation selectively in tumor cells [136].

Pyroptosis in tumors may not always be beneficial. For example, a recent study suggested that GSDME-mediated cancer cell pyroptosis can cause cytokine release syndrome (CRS), a complication associated with chimeric antigen receptor (CAR) T cell therapy [137]. In a mouse CAR T therapy model, massive B leukemic cell pyroptosis induced by CD19-recognizing CAR T cells resulted in a significant release of ATP, which in turn activated the NLRP3 inflammasome in host macrophages, thereby triggering CRS. CAR T-induced CRS was ameliorated by inhibition of the ATP-P_2_X_7_ pathway and caspase-1 activity. Although expression levels of GSDME and GSDMB were positively correlated with clinical benefit in several kinds of cancers, increased GSDMB expression was associated with poor clinical outcome in breast cancer [28,30,64,74,75]. Also, upregulation of GSDMC was observed in breast cancer, which was correlated with poor survival [29]. That is, GSDMs may also have pro-tumor effects. One possible explanation for this is that unlike acute pyroptosis induction, chronic induction of pyroptosis in tumors may lead to chronic inflammation, which creates a tumor-promoting microenvironment (Figure 5b). Gradual release of small amounts of ATP from tumor cells may affect antitumor immunity, as extracellular ATP can be rapidly broken down to adenosine, an immunosuppressive molecule [92]. Thus, it may be assumed that inhibition of GSDMs suppresses tumor progression in some cases. If one expects therapy-induced pyroptosis to improve the tumor microenvironment, it may be necessary to find the optimal extent of pyroptosis induction that should be neither too strong nor too weak. However, the possibility that GSDMB/C promote breast cancer through pyroptosis-independent mechanisms is not ruled out.

## 6. Conclusions

Recent studies have significantly advanced our knowledge of the mechanisms and pathophysiological roles of pyroptosis. GSDMs are activated by caspases and granzymes that are involved in the induction of apoptosis and similar cell death, and thus, pyroptosis and apoptosis seem to be two sides of the same coin (Figure 2). The molecular mechanisms of pyroptosis are also interesting from an evolutionary perspective, since GSDMs are a gene family that developed after vertebrates. Pyroptosis may change the tumor microenvironment from “cold” to “hot”, and the utilization of its effects, especially in combination with immunotherapy, are promising in terms of cancer treatment. However, the beneficial and detrimental effects of pyroptosis need to be further evaluated. In particular, the efficacy and limitations of pyroptosis-associated therapies need to be further investigated in clinical settings.

## Figures and Tables

**Figure 1 ijms-22-00426-f001:**
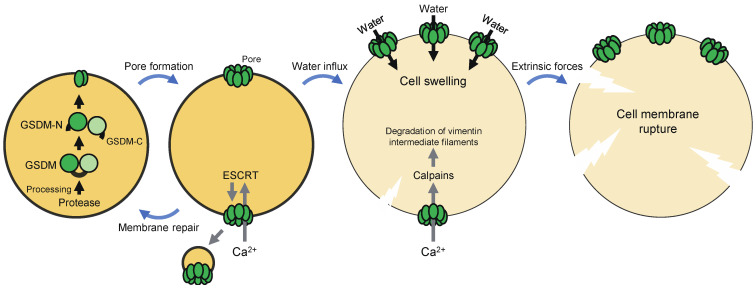
The molecular mechanism of pyroptosis. Pyroptosis is a necrotic form of regulated cell death executed by gasdermin (GSDM) family members. GSDMs, which are expressed as inactive forms, are proteolytically activated by certain proteases. The *N*-terminal fragments of GSDMs form pores in the plasma membrane. These pores cause water influx driven by osmotic pressure, leading to cell swelling and ultimately plasma membrane rupture. GSDM pores also allow the influx of Ca^2+^, which directs the assembly of the endosomal sorting complex required for transport (ESCRT) machinery. The ESCRT machinery mediates plasma membrane repair by promoting budding and pinching off of the damaged membrane. Calpains are also activated, which in turn degrade vimentin intermediate filaments, thereby enhancing rupture of the plasma membrane by extrinsic forces.

**Figure 2 ijms-22-00426-f002:**
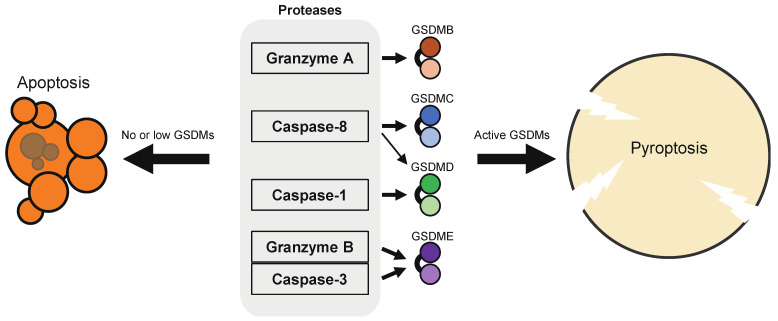
Proteases that activate GSDM members. GSDMB is processed by granzyme A into the active form; GSDMC by caspase-8; GSDMD by caspase-1, and to a lesser extent, by caspase-8; GSDME by caspase-3 and granzyme B. Of note, these GSDM-activating proteases can also induce or execute apoptosis or ROS-dependent, apoptosis-like cell death in cells that do not express sufficient levels of GSDMs for causing pyroptosis, in other words, caspase/granzyme-induced apoptosis can be switched to pyroptosis by the expression of GSDMs. GSDMD is also proteolytically activated by caspase-11/4/5, neutrophil elastase, and cathepsin G.

**Figure 3 ijms-22-00426-f003:**
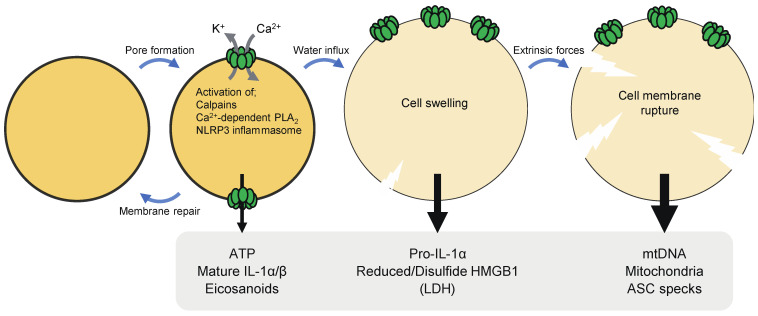
Damage-associated molecular patterns (DAMPs) and cytokines released from cells undergoing pyroptosis. The formation of GSDM pores in the plasma membrane causes the dissipation of ion gradients and release of small cytosolic molecules that can pass through the pores. Thereafter, GSDM pores cause cell lysis due to water influx, allowing the release of large cytosolic contents. ATP and mature IL-1α/β can be released through GSDM pores. The release of HMGB1 appears to require cell lysis. Immunostimulatory forms of (reduced and disulfide) HMGB1 may be released from pyroptotic cells. Severe cell membrane rupture can lead to the release of mitochondrial DNA (mtDNA), mitochondria, nuclei, and caspase recruitment domain (ASC) specks. Ca^2+^ influx via GSDM pores results in the activation of calpains, which process pro-IL-1α into the mature form. Ca^2+^-dependent phospholipase A_2_ (PLA_2_) also produces free arachidonic acid, which is the precursor for other eicosanoids. On the other hand, K^+^ efflux via GSDMD pores activates the NLRP3 inflammasome.

**Figure 4 ijms-22-00426-f004:**
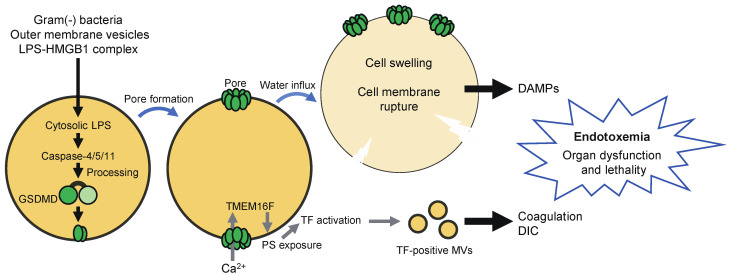
Pyroptosis contributes to the pathology of endotoxemia. Infection with Gram-negative bacteria may result in the release of lipopolysaccharide (LPS) into the cytoplasm of host cells. Outer membrane vesicles and HMGB1 can also deliver LPS into the cytoplasm. Cytoplasmic LPS then activates caspase-11/4/5, leading to proteolytic activation of GSDMD and consequent pyroptosis. DAMPs released from pyroptotic cells are thought to contribute to the pathology of endotoxemia. In addition, Ca^2+^ influx via GSDMD pores triggers the activation of tissue factor (TF) through TMEM16F-mediated phosphatidylserine (PS) exposure. Tissue factor-containing microvesicles (MVs) induce coagulation and play a critical role in the development of disseminated intravascular coagulation (DIC), which is associated with increased mortality in septic patients.

**Figure 5 ijms-22-00426-f005:**
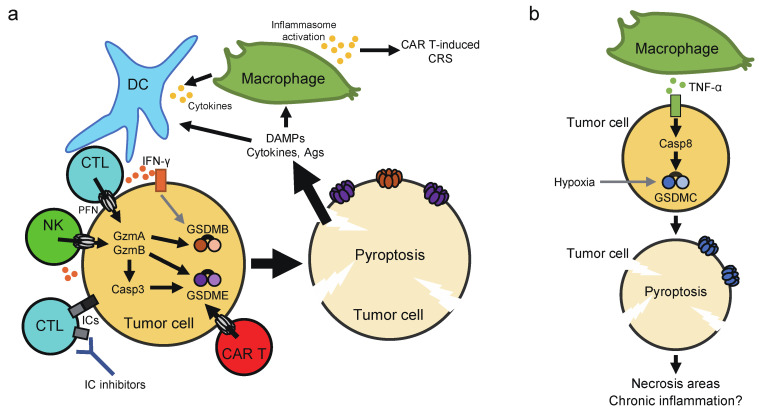
Effects of pyroptosis on the tumor microenvironment. (**a**) With the aid of perforin (PFN), Granzyme A/B (GzmA/B) are delivered from cytotoxic lymphocytes into the cytoplasm of target cancer cells. These Gzms activate GSDMB/E directly or by inducing caspase-3 (Casp3) activation, leading to tumor cell pyroptosis. Pyroptotic tumor cells also appear to activate the tumor microenvironment toward an immunostimulatory state probably through release of DAMPs and cytokines, forming a positive feedback loop in antitumor immunity. These effects of pyroptosis may synergize with immune checkpoint (IC) inhibitors. GSDMB expression is upregulated by IFN-γ. Massive tumor cell pyroptosis induced by CAR T therapy may cause CRS, which depends on a significant release of ATP and consequent activation of the NLRP3 inflammasome in host macrophages. (**b**) The expression of GSDMC is increased by hypoxia. Macrophage-derived TNF-α induces pyroptosis in tumor cells through the caspase-8-GSDMC pathway. This may account for the formation of necrotic areas found in central hypoxic regions of solid tumors. The sustained induction of pyroptosis in a small population of tumor cells may also induce chronic inflammation that promote tumor progression.

**Figure 6 ijms-22-00426-f006:**
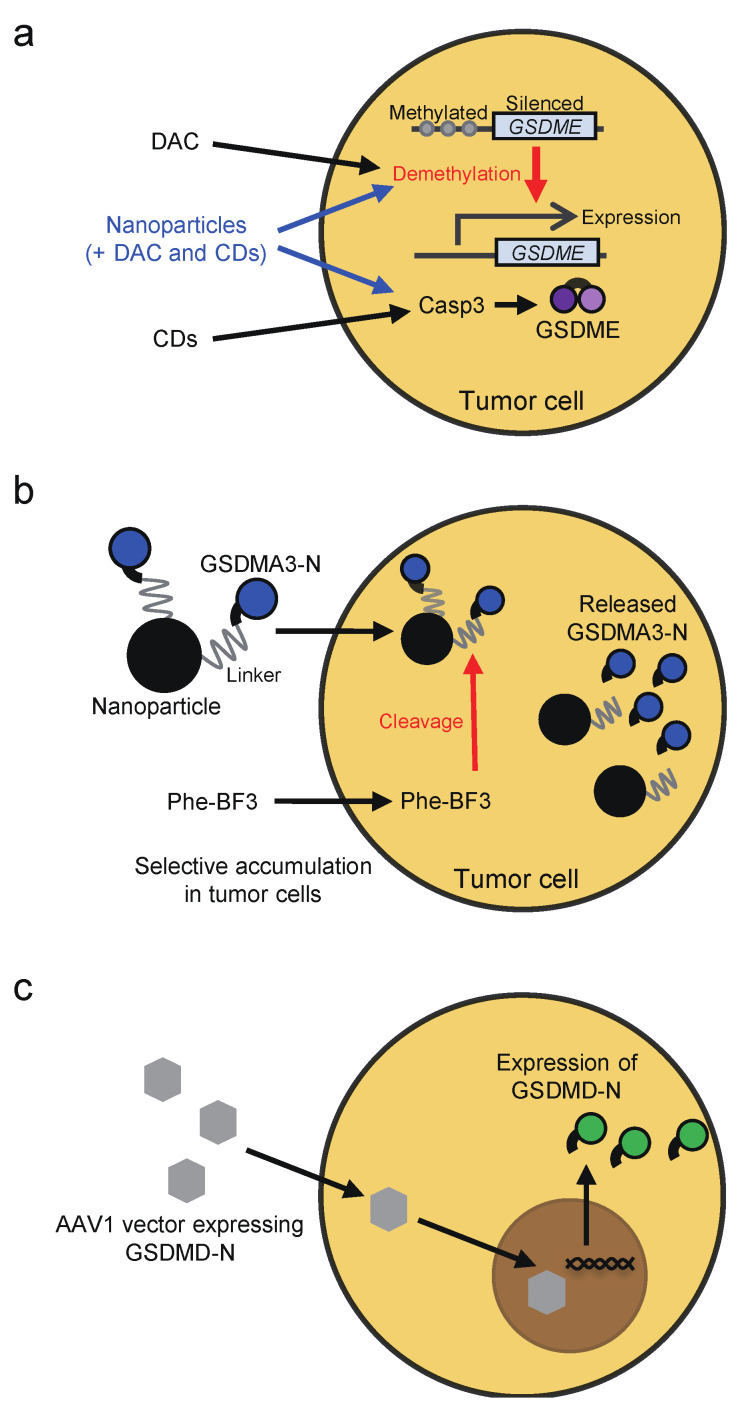
Strategies to induce pyroptosis in tumor cells. (**a**) GSDME expression is silenced in most cancer cells, likely attributed to promoter methylation. Treatment with the DNA methyltransferase inhibitor decitabine (DAC) restores GSDME expression, thereby sensitizing cancer cells to pyroptosis. The combination of decitabine and chemotherapeutic drugs (CDs) can lead to tumor cell pyroptosis via the caspase-3-GSDME pathway. Tumor-targeting nanoparticles loaded with decitabine and chemotherapeutic drugs may be useful to induce pyroptosis in tumors without causing damage in normal tissues. (**b**) A bioorthogonal chemical system has been established to induce tumor cell pyroptosis. In this system, active GSDMA3 protein is conjugated to gold nanoparticles via a silyl ether linker that can be cleaved by desilylation that is catalyzed by phenylalanine trifluoroborate (Phe-BF3), a cancer-imaging probe. Both the nanoparticles and Phe-BF3 selectively accumulate in tumors, enabling controlled release of active GSDMA3 in tumor cells. (**c**) Viral transduction of active forms of GSDMs can also lead to pyroptosis. In a previous study, adeno-associated serotype 1 virus (AAV1)-based vector that expresses GSDMD-N under the control of the schwann-cell-specific promoter P0 was made for the purpose of inducing pyroptosis in schwannomas.

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
