# Peer review of "Switching from Apoptosis to Pyroptosis: Gasdermin-Elicited Inflammation and Antitumor Immunity"

_ijms, 2021, doi:10.3390/ijms22010426_

Round 1

Reviewer 1 Report

COMMENTS IJMS-1038638:

In this review paper, the authors summarize recent studies about mechanisms and pathophysiological roles of pyroptosis and the importance of GSDMs in this process. They also presented evidence as to how GSDMs are activated by caspases and granzymes, most of which can also induce apoptosis in different situations. In addition, the authors show evidence that demonstrated that pyroptosis appears to facilitate the killing of tumor cells by cytotoxic lymphocytes, and it may also reprogram the tumor microenvironment to an immunostimulatory state. The better understanding of pyroptosis may help the development of cancer immunotherapy. I have considered the study very interesting. However, some issues have to be addressed to make the study easy to follow. 

 Comment 1. I have considered that the abstract is not consistent with the title, because the authors did not mention about the switching from apoptosis to pyroptosis.

Comment 2.  In the introduction, the authors mentioned that several modes of regulated necrosis have been reported, for example pyroptosis, necroptosis, and ferroptosis, which are induced by different stimuli. However, they just explain two of them. They have to briefly mention the ferroptosis process.

Comment 3.  The authors have to mention when the pyroptosis term was established, between the lines 68-70.

Comment 4.  Missing a reference in line 89.  

Comment 5. The authors have to mention the evidence about how the cells do not express sufficient levels of GSDMD.

Comment 6. What PRRs stands for.

Comment 7. It will be very helpful if the authors present a scheme to show how Caspase-11/4/5 contribute to the pathology of endotoxemia through the induction of pyroptosis.

Comment 8. There is a missing connection between point 4 and 4.1.

Comment 9. Fix “Po-IL-1β” in line 373.

Comment 10. I considered that the title is not appropriate because the manuscript is not really focusing in the switch from apoptosis to pyroptosis.

Author Response

First of all, I would like to thank the referee for the comments, which helped me to revise the manuscript. I have addressed all the comments as much as possible. Point-by-point responses to the comments are shown below. I appreciate your consideration and hope that these revisions are sufficient to make this manuscript suitable for publication.

Comment 1. I have considered that the abstract is not consistent with the title, because the authors did not mention about the switching from apoptosis to pyroptosis.

Reply: I think the title “Switching from Apoptosis to Pyroptosis” is a very accurate description of the properties of the GSDM family proteins. I emphasized (especially in Figure 2) that caspases/granzymes that activate GSDMs can also induce or execute apoptosis in the absence of the corresponding GSDMs; that is, caspase/granzyme-induced apoptosis can be switched to pyroptosis by the expression of GSDMs. To address this comment, I revised the Abstract as follows:

“GSDMs are activated by caspases and granzymes, most of which can also induce apoptosis in different situations, for example where the expression of GSDMs is too low to cause pyroptosis; that is, caspase/granzyme-induced apoptosis can be switched to pyroptosis by the expression of GSDMs.” (lines 18-20).

Comment 2.  In the introduction, the authors mentioned that several modes of regulated necrosis have been reported, for example pyroptosis, necroptosis, and ferroptosis, which are induced by different stimuli. However, they just explain two of them. They have to briefly mention the ferroptosis process.

Reply: To address this comment, I have added the following sentence to the text.

 “Ferroptosis involves abnormalities in lipid peroxidation metabolism and iron homeostasis.” (line 57)

Comment 3.  The authors have to mention when the pyroptosis term was established, between the lines 68-70.

Reply: To address this comment, the text has been modified as follows:

 “Pyroptosis was originally proposed by Cookson and Brennan as caspase-1-dependent non-apoptotic cell death induced in macrophages during infection with Salmonella Typhimurium.[6][7] In 2001, these researchers coined the term “pyroptosis” that stems from the Greek roots “pyro”, relating to fire or fever, and “ptosis” to denote a falling, to describe pro-inflammatory regulated necrosis.[7]” (lines 68-71)

Comment 4.  Missing a reference in line 89.  

Reply: I have added a citation to the description. (line 91)

Comment 5. The authors have to mention the evidence about how the cells do not express sufficient levels of GSDMD.

Reply: In previous reports, caspase-1-dependent apoptosis occurred in neuronal cells and mast cells, and GSDMD expression was significantly lower in these cell types than in macrophages. Therefore, it appears that there are cell types that do not express sufficient levels of GSDMD for causing pyroptosis and undergo apoptosis following caspase-1 activation. To address this comment, the text has been modified as follows:

“On the other hand, caspase-1 initiates apoptosis in cell types that do not express sufficient levels of GSDMD, such as neuronal cells and mast cells.[5][35][36]” (lines 168-169)

Comment 6. What PRRs stands for.

Reply: PRRs stands for “pattern recognition receptors”. Please see the Abbreviations section.

Comment 7. It will be very helpful if the authors present a scheme to show how Caspase-11/4/5 contribute to the pathology of endotoxemia through the induction of pyroptosis.

Reply: To address this comment, I have added Figure 4 and its legend to the manuscript. (lines 503-519)

Comment 8. There is a missing connection between point 4 and 4.1.

Reply: To address this comment, I have added a sentence to the text as follows:

“In the following subsections, individual DAMPs released from pyroptotic cells are described.” (lines 354-355)

Comment 9. Fix “Po-IL-1β” in line 373.

Reply: I am sorry for the mistake and thank you for pointing out the typo. I have corrected it. (line 387)

Comment 10. I considered that the title is not appropriate because the manuscript is not really focusing in the switch from apoptosis to pyroptosis.

Reply: As mentioned in my reply to the comment 1, I think the title “Switching from Apoptosis to Pyroptosis” is a very accurate description of the properties of the GSDM family proteins, and this point is  emphasized in this review article (especially in Figure 2). Hence, the title is not changed. Instead, the Abstract and legend of Fig. 2 were modified. (lines 18-20, lines 153-154)

Reviewer 2 Report

In the manuscript entitled “Switching from Apoptosis to Pyroptosis: Gasdermin-Elicited Inflammation and Antitumor Immunity”, a thorough review of the past and current reports on apoptosis, pyroptosis and the role of several mediator molecules especially Gasdermins are presented. The author convincingly presented molecular findings that link these cell death pathways with each other and also with anti-tumor immunity. The review is well written and highlights key molecular functions of the Gasdermin proteins, their expression patterns as well as regulation in normal and malignant cells and also the roles they play in the regulation of the tumor microenvironment and antitumor immunity. The review provides models that are easy to follow and represents a good guide for the development of therapies for the future management of cancers, as it summarizes and highlights the pros and cons of pyroptosis in cancer and also their therapy related counter effects.

General question: What is the role of pyroptosis in thermal therapy? Are there any effects on Gasdermins? The review would be very appealing to readers if examples on this are included. 

Minor corrections recommended as follows:

 1.) The abbreviation list is complete and arranged alphabetically to ease search.

2.) Check out and correct typos as follows:

  • Line 39: “For these reasons…” instead of “For the…”
  • Line 48: change “… mentioned above properties” to “…above mentioned properties”
  • Line 79: change “… is consisting of…” to “… consists of…”
  • Line 134: Pyroptosis instead of pyrotosis.
  • Line 279: do you mean “… role in asthma pathogenesis”?
  • Line 373: correct Pro-IL-1ß
  • Line 403: “…eventhough…” instead of even.
  • Line 412: should read “… GSDM pores are not...”. right?
  • Line 421: correct “CXCL12” in the brackets.
  • Line 598: correct “… expressing..”
  • Line 647: correct “…scan…” to “…can…”

3.) The sentence in Lines 471-473 is confusing. Something’s lacking.

Author Response

First of all, I would like to thank the referee for the comments, which helped me to revise the manuscript. I have addressed all the comments as much as possible. Point-by-point responses to the comments are shown below. I appreciate your consideration and hope that these revisions are sufficient to make this manuscript suitable for publication.

General question: What is the role of pyroptosis in thermal therapy? Are there any effects on Gasdermins? The review would be very appealing to readers if examples on this are included. 

Reply: Unfortunately, the involvement of pyroptosis and GSDM proteins in thermal therapy has not been established well. A previous paper suggested that magnetic intra-lysosomal hyperthermia might induce caspase-1-depedent pyroptosis in tumor cells. I added a sentence to the text to introduce this study, as follows:

“Furthermore, caspase-1-dependent pyroptosis occurred in INR1G9 tumor cells upon magnetic intra-lysosomal hyperthermia generated by the combination of targeted magnetic nanoparticles and an alternating magnetic field through leakage of lysosomal cathepsins into the cytosol, providing a strategy to induce caspase-1 activation selectively in tumor cells.” (lines 636-640)

Minor corrections recommended as follows:

 1.) The abbreviation list is complete and arranged alphabetically to ease search.

Reply: Thank you for pointing out this. In the current manuscript, the abbreviation list is arranged alphabetically.

2.) Check out and correct typos as follows:

  • Line 39: “For these reasons…” instead of “For the…”
  • Line 48: change “… mentioned above properties” to “…above mentioned properties”
  • Line 79: change “… is consisting of…” to “… consists of…”
  • Line 134: Pyroptosis instead of pyrotosis.
  • Line 279: do you mean “… role in asthma pathogenesis”?
  • Line 373: correct Pro-IL-1ß
  • Line 403: “…eventhough…” instead of even.
  • Line 412: should read “… GSDM pores are not...”. right?
  • Line 421: correct “CXCL12” in the brackets.
  • Line 598: correct “… expressing..”
  • Line 647: correct “…scan…” to “…can…”

Reply: Thank you for pointing out the typos. I have corrected all of them.

3.) The sentence in Lines 471-473 is confusing. Something’s lacking.

Reply: Different properties of PGE2 found in different studies are described in these sentences. I have modified the sentences to make it easier to understand, as follows:

“Eicosanoids are also involved in neutrophil recruitment to the PITs.[117] On the other hand, a study has suggested that PGE2 also serves as an inhibitory DAMP, as this lipid mediator in necrosis cell supernatants inhibited inflammatory responses to the supernatants.[118]” (lines 488-490)